# Assessment of Nitrate Removal Capacity of Two Selected Eukaryotic Green Microalgae

**DOI:** 10.3390/cells10092490

**Published:** 2021-09-20

**Authors:** Vaishali Rani, Gergely Maróti

**Affiliations:** 1Faculty of Science and Informatics, University of Szeged, 6720 Szeged, Hungary; ranivais@brc.hu; 2Biological Research Centre, Institute of Plant Biology, 6726 Szeged, Hungary

**Keywords:** nitrate, microalgae, lipids, *Chlorella*, *Chlamydomonas*

## Abstract

Eutrophication is a leading problem in water bodies all around the world in which nitrate is one of the major contributors. The present study was conducted to study the effects of various concentrations of nitrate on two eukaryotic green microalgae, *Chlamydomonas* sp. MACC-216 and *Chlorella* sp. MACC-360. For this purpose, both microalgae were grown in a modified tris-acetate-phosphate medium (TAP-M) with three different concentrations of sodium nitrate, i.e., 5 mM (TAP-M5), 10 mM (TAP-M10) and 15 mM (TAP-M15), for 6 days and it was observed that both microalgae were able to remove nitrate completely from the TAP-M5 medium. Total amount of pigments decreased with the increasing concentration of nitrate, whereas protein and carbohydrate contents remained unaffected. High nitrate concentration (15 mM) led to an increase in lipids in *Chlamydomonas* sp. MACC-216, but not in *Chlorella* sp. MACC-360. Furthermore, *Chlamydomonas* sp. MACC-216 and *Chlorella* sp. MACC-360 were cultivated for 6 days in synthetic wastewater (SWW) with varying concentrations of nitrate where both microalgae grew well and showed an adequate nitrate removal capacity.

## 1. Introduction

Increasing anthropogenic pressure on the water bodies has led to the problem of eutrophication all over the world, in which nitrate has emerged as one of the major pollutants [1]. This eutrophication, resulting from nutrient enrichment of nitrogen and phosphorus, poses a major threat to the aquatic ecosystem. The major factors behind eutrophication are the extensive use of fertilizers in agricultural fields and improper disposal of wastewater in the water bodies. Eutrophication causes a decrease in macrophyte abundance, an increase in the growth of algae and planktons, algae blooms and deoxygenation [2,3]. The World Health Organization and European Drinking Water Directive have set the limit of 50 mg NO_3_^-^ L^−1^ in drinking water to prevent the adverse effects of nitrate on human health [4].

Algae are the primary photosynthesizers present in the ecosystem and can be unicellular or multicellular. They can be found anywhere, from common environments, such as marine water and freshwater, to extreme environments, such as deserts, arctic, hypersaline habitats, etc. [5,6]. Nitrogen is one of the most important nutrients for algae growth and can be obtained from both organic (urea and amino acids) or inorganic (nitrate, nitrite and ammonia) sources. Microalgae are capable of increasing dissolved oxygen in the culture, as well as utilizing nutrients and carbon dioxide, thereby giving a protein-, carbohydrate- and lipid-rich algal biomass which can be further used for the production of biofuels, agricultural fertilizers, animal feedstock, etc. [7].

Domestic and industrial sewage contain high concentrations of nitrogen, phosphorus and organic matter in both soluble and particulate form. Due to their ability to utilize nitrogen and phosphorus, microalgae are gaining attention for the treatment of wastewater. This eco-friendly treatment consumes less energy, significantly reduces carbon emissions and can lead to the production of biofuels [8]. Furthermore, recovered nitrogen- and phosphorus-rich algal biomass can be exploited as low-cost fertilizer or as animal feed [9,10]. Several microalgae, namely, *Nannochloropsis oceanica*, *Nannochloropsis oculata, Scenedesmus* sp., *Demodesmus abundans*, *Chlorella vulgaris, Chlamydomonas reinhardtii* and *Chlorella* sp. have been studied for nitrogen removal [8,11,12,13,14]. *Chlamydomonas reinhardtii* have been shown to remove nitrogen at the rate of 55.8 mg L^−1^ day^−1^ from the wastewater cultivated in a biocoil with a high dry biomass yield [15]. In another study, it was shown that *Neochloris oleoabundans* can remove nitrate at the rate of 43.7 mg L^−1^ day^−1^ from the artificial wastewater containing 140 mg N-NO_3_^-^ up to a near-zero residue nitrate level [16]. Moreover, research have been going on to utilize algal–bacterial interactions for wastewater treatment [9,17].

Multiple factors can influence photosynthesis, biomass production, biochemical and physiological composition of microalgae. Light conditions, temperature, pH, nutrient supply and salinity are among the most important parameters. Nitrogen is one of the key nutrients to the algae and a change in its level can affect the growth rate, lipid content, carbohydrate content and protein content of the microalgae. Several studies have shown that nitrogen limitation enhances the production of lipids and carbohydrates in microalgae at the cost of low biomass productivity and lowered growth rate [18,19,20]. Gour et al., showed, in their study, that lower nitrate concentrations lead to high lipid content and lipid productivity in *Scenedesmus dimorphus* [21]. In contrast, other studies have also shown an increase in the amount of lipids by increasing nitrate concentrations to a certain limit in microalgae *Chlorella* sp. and *Isochrysis galbana* [22,23]. Lipid content in *Chlorella minutissima* increased from 22.7% to 36% when the nitrate concentration increased from 57 mg L^−1^ to 225 mg L^−1^ [24]. Protein levels have shown to be increased from 16.87% to 47.75% with the increase in the concentration of nitrate from to 0 to 247 mg L^−1^ in *Scenedesmus* sp. CCNM 1077 [19]. In algae, chlorophyll a levels also seem to vary with the concentration of nitrate [22,23,25,26]. In *Ulva rigida* and *Neochloris oleoabundans*, chlorophyll a level increased as the concentration of nitrate was increased, but not all of the microalgae follow the same pattern; in some cases, the concentration of chlorophyll a decreased with the increasing nitrate concentration [23,25,26]. Another study has shown thathigh nitrate concentration leads to the production of sulfated polysaccharides with potent bioactive properties in *Chlamydomonas reinhardtii* [27].

In the current study, *Chlamydomonas* sp. MACC-216 and *Chlorella* sp. MACC-360 were investigated for their growth and nitrate removal properties on various concentrations of nitrate. Our study aimed to understand the influence of nitrate on the growth and to assess the nitrate removal capacity of the two selected microalgae using modified tris-acetate-phosphate (TAP) medium and synthetic wastewater (SWW). The effects of different nitrate concentrations on the accumulation of proteins, carbohydrates and lipids were also investigated in the microalgae.

## 2. Materials and Methods

### 2.1. Microalgae Strains and Growth Media

Two strains of microalgae were selected, namely, *Chlamydomonas* sp. MACC-216 and *Chlorella* sp. MACC-360, for the present study. These strains were provided by the Mosonmagyaróvár Algae Culture Collection (MACC). The TAP medium consisted of 2.42 g L^−1^ of Tris base, 0.374 g L^−1^ of NH_4_Cl, 0.204 g L^−1^ of MgSO_4_ 7H_2_O, 0.066 g L^−1^ of CaCl_2_ 2H_2_O, 0.287 g L^−1^ of K_2_HPO_4_, 0.142 g L^−1^ of KH_2_PO_4_, 0.049 g L^−1^ of Na_2_EDTA·2H_2_O, 0.039 g L^−1^ of ZnSO_4_·7H_2_O, 0.011 g L^−1^ of H_3_BO_3_, 0.007 g L^−1^ of MnCl_2_·4H_2_O, 0.008 g L^−1^ of FeSO_4_·7H_2_O, 0.002 g L^−1^ of CoCl_2_·6H_2_O, 0.002 g L^−1^ of CuSO_4_·5H_2_O, 0.001 g L^−1^ of (NH_4_)_6_Mo_7_O_24_·4H_2_O and 1 mL L^−1^ of CH_3_COOH and the pH was maintained at 7. The final concentration of CH_3_COOH in the TAP medium was 16.8 mM. To study the effects of nitrate on the microalgae, the TAP medium was modified by substituting sodium nitrate as the nitrogen source (TAP-M) instead of ammonium chloride. In addition, 0.001 g L^−1^ of (NH_4_)_6_Mo_7_O_24_·4H_2_O was replaced with 0.006 g L^−1^ of Na_2_MoO_4_·2H_2_O in the modified TAP medium. First, screening was performed for the selection of nitrate concentrations to be used in further experiments. The growth of both microalgae was tested in TAP-M containing 1 mM (84.99 mg L^−1^), 5 mM (424.97 mg L^−1^), 10 mM (849.94 mg L^−1^), 15 mM (1.27 g L^−1^), 20 mM (1.69 g L^−1^), 40 mM (3.39 g L^−1^), 50 mM (4.24 g L^−1^), 75 mM (6.37 g L^−1^) and 100 mM (8.49 g L^−1^) nitrate. Three different concentrations of sodium nitrate (5 mM, 10 mM and 15 mM) were selected for further experiments. Both microalgae were cultivated in TAP and TAP-M with 5 mM (TAP-M5), 10 mM (TAP-M10) and 15 mM (TAP-M15) concentrations of sodium nitrate at 25 °C under a light intensity of 50 µmol m^−2^ s^−1^ with continuous shaking at 180 rpm in a regime of 16:8 light–dark periods.

### 2.2. Growth Parameters

*Chlamydomonas* sp. MACC-216 and *Chlorella* sp. MACC-360 were grown in each TAP, TAP-M5, TAP-M10 and TAP-M15 media in two separate 24-well plates. The initial absorbance at 720 nm (day 0) for both microalgae in all four media was kept at 0.1. Absorbance was measured daily for 6 days at 720 nm for both microalgae in a Hidex microplate reader. For cell counting, a LUNA cell counter was used which counted the number of cells on the basis of autofluorescence emitted by microalgae. For cell size, samples of both microalgae were collected from 3-day old cultures and microalgae were observed under an Olympus Fluoview FV1000 confocal laser scanning microscope. Images were taken with a 60× magnification objective and cell perimeter was calculated using ImageJ.

The growth patterns of both microalgae were determined by their number of generations (n) and mean generation time per day (g) in the logarithmic growth phase according to the following equations [28]:(1)$$n=\frac{{\mathrm{logN}-\mathrm{logN}}_{0}}{\log2}$$
(2)$$g=\frac{t}{n}$$
where ‘n’ is the number of generations in a given time period, ‘N_0_′ and ‘N’ are the initial and final cell number of microalgae, ‘g’ is the mean generation time, and ‘t’ is the duration of the exponential growth phase. The specific growth rate (day^−1^) ‘µ’ was also calculated for both microalgae.
(3)$$\mu =\frac{\ln2}{g}$$

### 2.3. Nitrate Determination by the Salicylic Acid Method

For nitrate removal experiments, *Chlamydomonas* sp. MACC-216 and *Chlorella* sp. MACC-360 were grown in each TAP-M5, TAP-M10 and TAP-M15 in two separate 24-well plates. The initial absorbance at 720 nm (day 0) for both microalgae in all three media was kept at 0.1. Nitrate removal was determined from day 0 to day 6 in TAP-M5, TAP-M10 and TAP-M15 media for both microalgae. To calculate the nitrate removal rate, first, both microalgae were cultivated in 20 mL of TAP medium for 3 days; then, on the 3rd day, cultures of both microalgae were centrifuged at 4000 rpm for 10 min and then washed with fresh TAP-0 medium (TAP without any nitrogen source). After washing, both cultures were divided and re-suspended into TAP-M5, TAP-M10 and TAP-M15 media. The nitrate removal rate was determined every 3 h for up to 9 h. For the analysis of nitrate removal and removal rate, a nitrate assay was performed as described by Cataldo et al. [29]. Briefly, 10 µL of the sample was taken in a microcentrifuge tube and 40 µL of 5% (*w*/*v*) salicylic acid in concentrated H_2_SO_4_ was slowly added to the tube and mixed properly. After 20 min of incubation at room temperature, 950 µL of 2M NaOH was slowly added to the tube and mixed. The sample was cooled down to room temperature and absorbance was determined at 410 nm in a Hidex microplate reader.

### 2.4. Determination of Reactive Oxygen Species (ROS)

ROS production was measured by 2′,7′-dichlorodihydrofluorescein diacetate (DCFH-DA) as described by Wang et al. [30]. The stock solution of DCFH-DA was prepared in DMSO at a final concentration of 10 mM and stored at −20 °C until further use. For the determination of ROS, 3-day old cultures of both microalgae grown in TAP media were harvested by centrifugation at 4000 rpm for 10 min. The pellets were washed once with 1X phosphate-buffered saline (PBS) (pH of 7.0) followed by resuspension in 1× PBS. Both microalgae cultures were incubated at 25 °C in a shaker incubator for one hour in the dark. After 1 h, both cultures were centrifuged and washed, followed by division and resuspension into TAP, TAP-M5, TAP-M10 and TAP-M15 media containing 5 µM DCFH-DA. Resuspension was carried out in 48-well plates. Separate plates were used for *Chlamydomonas* sp. MACC-216 and *Chlorella* sp. MACC-360. For blank, only respective media with 5 µM DCFH-DA were used and the blank measurement was carried out in a separate 48-well plate. All of the plates were incubated at 25 °C in a shaker incubator under constant illumination. The measurements for ROS production were conducted every hour for up to 4 h. The fluorescence of fluorescent 2′,7′-dichlorofluorescein (DCF) was measured in a Hidex microplate reader with excitation and emission filters set at 490 nm and 520 nm, respectively.

### 2.5. Total Pigments Extraction and Quantification

*Chlamydomonas* sp. MACC-216 and *Chlorella* sp. MACC-360 were grown in 10 mL of each TAP, TAP-M5, TAP-M10 and TAP-M15 for three days. For chlorophyll extraction, 10 mL culture of each 3-day old culture was taken and centrifuged at 4000 rpm for 10 min. The supernatants were discarded and then 5 mL of methanol was added to the pellets and mixed with pipetting. Then, the tubes were kept in the dark at 45 °C for 30 min. Afterwards, the samples were centrifuged at 8000 rpm for 10 min and supernatants were collected for absorbance. Absorbance was taken at 653 nm, 666 nm and 470 nm in a Hidex microplate reader. Calculations for chlorophyll a, chlorophyll b and total carotenoids were performed as described by Lichtenthaler and Wellburn [31].
C_a_ = 15.65A_666_ − 7.34A_653_(4)
C_b_ = 27.05A_653_ − 11.21A_666_
(5)
C_x+c_ = 1000A_470_ − 2.86C_a_ − 129.2C_b_/245(6)
where C_a_, Cb, Cx_+c_ are the amounts of chlorophyll a, chlorophyll b and total carotenoids, respectively, in µg mL^−1^.

### 2.6. Total Carbohydrates Extraction and Quantification

For total carbohydrates extraction, pellets obtained after total pigments extraction were used. The pellets were washed with Milli-Q water and then further dissolved in 10mL of Milli-Q. A volume of 1 mL from each dissolved pellet was taken in a fresh glass tube and 5 mL of anthrone reagent was added to it. The anthrone reagent was prepared freshly by dissolving 0.5 g of anthrone in 250 mL of concentrated sulfuric acid. After the addition of the anthrone reagent, tubes were cooled down and then incubated at 90 °C for 17 min in a water bath. After incubation, tubes were cooled down again to room temperature and the absorbance was taken at 620 nm in a Hidex microplate reader.

### 2.7. Total Proteins Extraction and Quantification

For protein extraction, *Chlamydomonas* sp. MACC-216 and *Chlorella* sp. MACC-360 were grown in 10 mL of each TAP, TAP-M5, TAP-M10 and TAP-M15 media for three days. A volume of 10 mL of each 3-day old culture was centrifuged at 4000 rpm for 10 min and the pellets were resuspended in 1 mL of lysis buffer. Working concentration of lysis buffer consisted of 50 mM Tris-Cl (pH of 8.0), 150 mM NaCl, 1 mM EDTA (pH of 8.0), 10% Glycerol and 1× cOmplete^TM^ EDTA-free protease inhibitor cocktail (Roche). After resuspension, sonication was carried out at 0.8 cycle, 90% amplitude for 10 min. After sonication, centrifugation was performed at 17,000 rpm for 20 min at 4 °C. The supernatants were collected in fresh microcentrifuge tubes and used for the Bradford assay. For the Bradford assay, samples were diluted 10 times; 50 µL of each diluted sample was added into a fresh microcentrifuge tube, followed by the addition of 1.5 mL of Bradford reagent. The samples were incubated for 10 min at room temperature and then the absorbance was measured at 595 nm in a Hidex microplate reader.

### 2.8. Total Lipids Extraction and Quantification

*Chlamydomonas* sp. MACC-216 and *Chlorella* sp. MACC-360 were grown in 10 mL of each TAP, TAP-M5, TAP-M10 and TAP-M15 for three days. For lipid extraction, 3-day old cultures of both microalgae were centrifuged at 4000 rpm for 10 min and the supernatants were discarded. Pellets were dissolved in 3 mL of chloroform: methanol (2:1). Dissolved pellets were sonicated at 90% amplitude for 2 min. After sonication, 2 mL of chloroform: methanol (2:1) was added to the mixture and incubation was carried out at room temperature for 1 h with constant shaking. After one hour, centrifugation was performed at 4000 rpm for 10 min. Supernatants were collected in fresh glass tubes and pellets were re-dissolved in 2 mL of chloroform: methanol (2:1) followed by further incubation at room temperature for half an hour with constant shaking. After half an hour, centrifugation was carried out at 4000 rpm for 10 min. Supernatants were collected in the tubes with previously isolated supernatants. To the supernatants, 1/5th volume of 0.9% NaCl was added and centrifugation was carried out at 4000 rpm for 10 min. Lower phases were collected in fresh glass tubes and were evaporated to collect total lipids. For total lipids quantification, first, phospho-vanillin reagent was prepared by dissolving 0.6 g of vanillin in 100 mL of hot water, followed by the addition of 400 mL of 85% phosphoric acid. This reagent can be stored in the dark for several months until it turns dark. Collected total lipids fractions were dissolved in 1 mL of chloroform; 100 µL from each total lipid fractions was taken in a fresh glass tube and evaporated at 90 °C in the water bath. A volume of 100 µL of concentrated sulfuric acid was added to each tube and tubes were incubated at 90 °C for 15 min in a water bath. Tubes were cooled down on the ice for 5 min and 2.4 mL of phospho-vanillin reagent was added to all the lipid samples, followed by incubation at room temperature for 5 min. Absorbance was measured at 530 nm in a Hidex microplate reader.

### 2.9. Confocal Microscopy with BODIPY Dye

*Chlamydomonas* sp. MACC-216 and *Chlorella* sp. MACC-360 were grown in TAP, TAP-M5, TAP-M10 and TAP-M15 in 24-well plates for three days. For the localization of lipids inside microalgae cells, BODIPY dye (Sigma-Aldrich, Burlington, MA, USA) was used. The stock solution of 4 mM BODIPY was prepared in 100% methanol. To 50 µL of 3-day old microalgae cells, 0.25 µL of 4 mM BODIPY dye was added and then microalgae were observed under an Olympus Fluoview FV1000 confocal laser scanning microscope. For BODIPY, the emission range was selected from 500 nm to 515 nm and images were taken by a 60X magnification objective at 6X zoom for *Chlamydomonas* sp. MACC-216 and 8X zoom for *Chlorella* sp. MACC-360.

### 2.10. Synthetic Wastewater Treatment

Synthetic wastewater (SWW) was prepared according to the procedure mentioned in “OECD guidelines for testing chemicals” [32]. Volumes of 16 g of peptone, 11 g of meat extract, 4.25 g of sodium nitrate (NaNO_3_), 0.7 g of sodium chloride (NaCl), 0.4 g of calcium chloride dehydrate (CaCl_2_·2H_2_O), 0.2 g of magnesium sulphate heptahydrate (MgSO_4_·7H_2_O) and 2.8 g of anhydrous potassium monohydrogen phosphate (K_2_HPO_4_) were added to 1 L of distilled water and the pH of this medium was set at 7.5. The initial concentration of sodium nitrate in synthetic wastewater was 50 mM; therefore, three more dilutions were made, i.e., 5 mM, 10 mM and 25 mM. *Chlamydomonas* sp. MACC-216 and *Chlorella* sp. MACC-360 were grown in synthetic wastewater (with the above-mentioned nitrate concentrations) for six days. The cultivation of both microalgae was carried out in a Multi-Cultivator (Photon Systems Instruments, Drásov, Czech Republic) at a continuous illumination of 50 µmol m^−2^ s^−1^ intensity. For six days, growth and nitrate removal capacity were observed in both microalgae.

### 2.11. Statistical Analyses

Statistical analyses were performed using RStudio version 1.2.5019. All measurements were performed in triplicates. Mean and standard deviation values were calculated. The error bars in the figures depict standard deviations. Significant difference among the means was calculated using the Tukey’s test. The difference among means was considered to be significant at the value of *p* < 0.05.

## 3. Results

### 3.1. Influence of Nitrate on the Growth Parameters of Chlamydomonas sp. MACC-216 and Chlorella sp. MACC-360

Both *Chlamydomonas* sp. MACC-216 and *Chlorella* sp. MACC-360 were grown in modified TAP supplemented with various concentrations of nitrate as a sole nitrogen source for the growth, i.e., 1 mM, 5 mM, 10 mM, 15 mM, 20 mM, 40 mM, 50 mM, 75 mM and 100 mM (Appendix Aa,b).

When the microalgae were grown in TAP, TAP-M5, TAP-M10 and TAP-M15, the peak of the growth measured through absorbance at 720 nm was observed on the third day in each media (Figure 1). Similar results were observed in both microalgae when the growth was observed on the basis of cell density except for *Chlorella* sp. MACC-360, which showed maximum cell density on the fourth day when grown in normal TAP (Figure 2). The maximum cell density for *Chlamydomonas* sp. MACC-216 and *Chlorella* sp. MACC-360 in TAP was 1.4 × 10^7^ cells mL^−1^ and 4.3 × 10^7^ cells mL^−1^, respectively. Both microalgae also had high cell densities in TAP-M5 (Figure 2). The cell density for *Chlamydomonas* sp. MACC-216 and *Chlorella* sp. MACC-360 in TAP-M5 was 1.28 × 10^7^ cells mL^−1^ and 3.7 × 10^7^ cells mL^−1^, respectively. It was observed, from the growth curve and cell density, that the growth of *Chlamydomonas* sp. MACC-216 and *Chlorella* sp. MACC-360 decreased as the concentration of nitrate increased (Figure 1 and Figure 2).

The growth of *Chlorella* sp. MACC-360 started to decline in TAP-M5, TAP-M10 and TAP-M15 after the third day, whereas, in comparison, *Chlamydomonas* sp. MACC-216 could sustain in TAP-M5, TAP-M10 and TAP-M15 media even on the sixth day (Figure 2). Table 1 shows the growth parameters for both *Chlamydomonas* sp. MACC-216 and *Chlorella* sp. MACC-360 in four different media. It was found that the specific growth rate of *Chlamydomonas* sp. MACC-216 was similar among all of the four media, whereas, in the case of *Chlorella* sp. MACC-360, the highest specific growth rate was observed in TAP. In *Chlorella* sp. MACC-360, the specific growth rate decreased as the concentration of nitrate increased, whereas the specific growth rate of *Chlamydomonas* sp. MACC-216 seemed unaffected by different nitrate concentrations. An increase was observed in the cell size of both *Chlamydomonas* sp. MACC-216 and *Chlorella* sp. MACC-360 cells in the presence of nitrate (Table 1).

### 3.2. Nitrate Removal by Chlamydomonas sp. MACC-216 and Chlorella sp. MACC-360

Nitrate removal by *Chlamydomonas* sp. MACC-216 and *Chlorella* sp. MACC-360 in TAP-M5, TAP-M10 and TAP-M15 was investigated daily for a 6-day-long period. The removal of nitrate by both microalgae was fast in the first three days from TAP-M5, TAP-M10 and TAP-M15, but it slowed down after the 3rd day (Figure 3). Furthermore, it was also observed that *Chlamydomonas* sp. MACC-216 performed better in removing nitrate from TAP-M5, TAP-M10 and TAP-M15 than *Chlorella* sp. MACC-360 (Table 2).

Both microalgae removed 100% of nitrate from TAP-M5 by the 3rd day. *Chlamydomonas* sp. MACC-216 removed 84% of nitrate from TAP-M10 and 53% from TAP-M15, whereas *Chlorella* sp. MACC-360 removed 72% of nitrate from TAP-M10 and 51% from TAP-M15 by the 6th day. Thus, *Chlamydomonas* sp. MACC-216 and *Chlorella* sp. MACC-360 can remove approximately 8 mM and 7.5 mM nitrate, respectively, by the 6th day from TAP-M10 and TAP-M15. Nitrate removal rate by both microalgae was calculated and it was observed that *Chlamydomonas* sp. MACC-216 had a higher removal rate than *Chlorella* sp. MACC-360 from the tested media (Table 3). Also, the nitrate removal rate of *Chlorella* sp. MACC-360 was concentration-dependent, whereas *Chlamydomonas* sp. MACC-216 did not follow the same pattern (Table 3).

### 3.3. Nitrate Led to ROS Production in Chlorella sp. MACC-360

DCF fluorescence was used as a measure of ROS content in both microalgae. A time-dependent increase in DCF fluorescence was observed in both microalgae. *Chlamydomonas* sp. MACC-216 showed less DCF fluorescence in TAP-M (TAP-M5, TAP-M10 and TAP-M15) than in TAP, which indicates that nitrate did not cause any major stress to *Chlamydomonas* sp. MACC-216 (Figure 4a). *Chlorella* sp. MACC-360 showed a significant rise in DCF fluorescence when grown under TAP-M15, indicating that 15 mM nitrate caused significant stress to *Chlorella* sp. MACC-360 (Figure 4b).

### 3.4. Nitrate Affected Total Pigments Production

Total pigments were extracted from 3-day old cultures of both *Chlamydomonas* sp. MACC-216 and *Chlorella* sp. MACC-360. The amount of pigments decreased in both *Chlamydomonas* sp. MACC-216 and *Chlorella* sp. MACC-360 as the nitrate concentration increased (Figure 5). However, in the case of *Chlamydomonas* sp. MACC-216, the difference in the amount of pigments among different media was not found to be significant, whereas the decrease in the amount of chlorophyll a in *Chlorella* sp. MACC-360 was significant. *Chlamydomonas* sp. MACC-216 was shown to have a generally higher amount of pigments than *Chlorella* sp. MACC-360. Especially, chlorophyll b was present in a much lower quantity in *Chlorella* sp. MACC-360 in comparison to *Chlamydomonas* sp. MACC-216.

### 3.5. Effects of Nitrate on Total Protein and Carbohydrate Contents

*Chlamydomonas* sp. MACC-216 and *Chlorella* sp. MACC-360 showed an increase in protein content when the concentration of nitrate was increased from 5 mM to 10 mM and then it decreased when the concentration was further increased from 10 mM to 15 mM but this increase and decrease in protein content was not significant. *Chlamydomonas* sp. MACC-216 yielded a larger quantity of total proteins in comparison to *Chlorella* sp. MACC-360 (Figure 6a). Overall, no significant difference was observed among the total protein contents of *Chlamydomonas* sp. MACC-216 grown in TAP, TAP-M5, TAP-M10 and TAP-M15. Likewise, there was no significant increase or decrease in total protein contents among the four samples of *Chlorella* sp. MACC-360. Nitrate did not influence the amount of total carbohydrates neither in *Chlamydomonas* sp. MACC-216 nor in *Chlorella* sp. MACC-360. Similar to protein contents, no statistically significant difference was observed in total carbohydrate contents among TAP, TAP-M5, TAP-M10 and TAP-M15 samples of both microalgae (Figure 6b). *Chlorella* sp. MACC-360 did show a higher amount of carbohydrates than *Chlamydomonas* sp. MACC-216 (Figure 6b).

### 3.6. Lipid Content Increased by Nitrate in Chlamydomonas sp. MACC-216

Total lipid contents were checked to see whether nitrate affected lipid production in *Chlamydomonas* sp. MACC-216 and *Chlorella* sp. MACC-360. Two different methods were used to determine lipid accumulation in the selected microalgae. First, BODIPY dye was used for the labelling of neutral lipids inside the microalgae cells, while the second method was the quantification of extracted lipids using the phospho-vanillin reagent assay. Lipid content was estimated from 3-day old cultures of both microalgae. In *Chlamydomonas* sp. MACC-216, the total lipid content increased from 19.66 mg g^−1^ of fresh weight (FW) in the microalgae grown in TAP to 37.51 mg g^−1^ of FW in the microalgae grown in TAP-M15 (Figure 7). In *Chlorella* sp. MACC-360, microalgae grown in TAP-M5, TAP-M10 and TAP-M15 showed no significant difference among each other or TAP in total lipid contents (Figure 7). 

Through BODIPY staining, it was observed that the fluorescence of dye increased as the concentration of nitrate increased in *Chlamydomonas* sp. MACC-216, indicating the presence of an increased amount of lipids in the microalgae grown in TAP-15 (Figure 8). *Chlorella* sp. MACC-360 cells (~5–10 in 100 cells) started showing BODIPY fluorescence in the presence of nitrate, while, in the case of TAP, none of the cells showed BODIPY fluorescence (Figure 8).

### 3.7. Chlorella sp. MACC-360 and Chlamydomonas sp. MACC-216 Efficiently Removed Nitrate from Synthetic Wastewater

Synthetic wastewater (SWW) was prepared to determine the nitrate removal capacity of axenic *Chlamydomonas* sp. MACC-216 and *Chlorella* sp. MACC-360 in a wastewater model system. SWW media with different concentrations of nitrate (5 mM, 10 mM, 25 mM and 50 mM) were prepared to check the growth of microalgae species. It was observed that *Chlamydomonas* sp. MACC-216 grew better in 5 mM and 10 mM SWW than in 25 mM and 50 mM SWW (Figure 9a). *Chlamydomonas* sp. MACC-216 showed the least growth in 50 mM SWW. *Chlorella* sp. MACC-360 grew better in high nitrate concentrations i.e., 25 mM and 50 mM than in SWW with 5 mM and 10 mM nitrate (Figure 9b). 

Nevertheless, *Chlamydomonas* sp. MACC-216 performed better in nitrate removal than *Chlorella* sp. MACC-360. While *Chlamydomonas* sp. MACC-216 removed 34.6% of nitrate from 50 mM SWW by the 6th day, *Chlorella* sp. MACC-360 were able to remove 27.6% of total nitrate in the same period. In both algae species, total nitrate removal in a 6-day-long period increased as the concentration of nitrate increased in the SWW (Table 4).

## 4. Discussion

Our results demonstrated the effect of nitrate on the growth of microalgae *Chlamydomonas* sp. MACC-216 and *Chlorella* sp. MACC-360. We investigated the growth of both microalgae in TAP-M media containing nitrate concentration from 1 mM to 100 mM (Appendix A). At high concentrations, the growth was strongly affected, but microalgae still managed to grow. Similar to our observations, *Chlorella vulgaris* have been shown to grow at a concentration of 97 mM nitrate [33]. The specific growth rate of *Chlamydomonas* sp. MACC-216 remained unaffected, irrespective of different nitrate concentrations. *Chlorella* sp. MACC-360 showed a decrease in the specific growth rate with the increase in concentration of nitrate, which correlates with the study performed by Jeanfils et al., [33] where they observed a decrease in the growth of *Chlorella vulgaris* when the concentration of nitrate exceeded 12 mM. In contrast, other studies have shown an increase in the growth with an increase in the concentration of nitrate, but, in all of these studies, the concentration of nitrate was lower than 5 mM [12,34]. Furthermore, cell size also seemed to be influenced by the presence of nitrate. The cell size of both microalgae grown in TAP, in comparison to TAP-M, was smaller (Table 1). The results indicate an increase in the cell size of microalgae with an increase in the concentration of nitrate. It is not clear what led to this increase in cell size; it could be due to the accumulation of lipids inside the cells in the case of *Chlamydomonas* sp. MACC-216. 

We observed that both *Chlamydomonas* sp. MACC-216 and *Chlorella* sp. MACC-360 were able to remove 100% of nitrate from TAP-M5 by the 3rd day. This nitrate removal can be explained by previous studies, which have shown nitrate assimilation reactions in *Chlamydomonas reinhardtii*; it has been stated that nitrate is first reduced to nitrite in the cytoplasm by nitrate reductase, followed by its transfer to chloroplast, where it is further reduced to ammonia by nitrite reductase and this ammonia is then incorporated in carbon skeletons by the Glutamine Synthetase/Glutamine Oxoglutarate Aminotransferase (GS/GOGAT) cycle, which is responsible for the synthesis of glutamate [35,36,37]. For the nitrate reduction in the cytoplasm, first, nitrate needs to enter inside microalgae cells, a process which is carried out by nitrate transporters. Three different gene families (Nrt1, Nrt2 and Nar1) have been stated to encode putative nitrate/nitrite transporters in *Chlamydomonas* [35,38]. These three families code for both high and low affinity nitrate transporters. Our results of nitrate removal efficiency are consistent with the Su et al. [7] study, where they showed a 99% nitrate removal efficiency performed by *Chlamydomonas reinhardtii* and *Chlorella vulgaris* by the 4th and 6th day, respectively. The high uptake of nitrate by microalgae is important, because nitrate is a nitrogen source which is important for microalgae survival; without any nitrogen sources, there is a deprivation of electron acceptors, i.e., NADP+, which plays a basic role during photosynthesis [22]. We also observed that, in *Chlorella* sp. MACC-360, the nitrate removal rate was dependent on the concentration of nitrate present in the medium; however, this case was not observed in *Chlamydomonas* sp. MACC-216 (Table 2 and Table 3). Similar to our observations on the nitrate removal rate of *Chlorella* sp. MACC-360, Jeanfils et al. [33] also noted an increase in the nitrate uptake rate by *Chlorella vulgaris* with an increase in the concentration of nitrate from 2 mM to 29 mM. The reason behind this could be the dependence of the activity of nitrate reductase on the actual concentration of nitrate [39]. 

ROS, including O_2_^-^, OH^-^ and H_2_O_2_, are produced as the endpoint of metabolic pathways which take place in the mitochondria, chloroplasts, peroxisomes, endoplasmic reticulum, chloroplast, cell wall and plasma membrane in freshwater microalgae [40]. ROS are known to be produced in response to various environmental stresses, such as drought, heavy metals, high salt concentration, UV irradiation, extreme temperatures, pathogens etc. [41]. ROS, in high amounts, are toxic to cells and lead to oxidative damage, which further causes cell death. In our study, we demonstrated the formation of total ROS in the microalgae cells exposed to different concentrations of nitrate by measuring DCF fluorescence. This fluorescence is obtained when cell-permeable indicator DCFH-DA is hydrolyzed by cellular esterases to form the non-fluorescent 2′,7′-dichlorodihydrofluorescein (DCFH), which is further transformed to highly fluorescent 2′,7′-dichlorofluorescein (DCF) in the presence of ROS. Only *Chlorella* sp. MACC-360 grown in TAP-M15 media showed the highest ROS production between studied microalgae, which points toward the significant stress caused to *Chlorella* sp. MACC-360 by high concentration of nitrate. 

The abundance of photosynthetic pigments (chlorophyll a, b and carotenoids) decreased as the concentration of nitrate increased in both *Chlamydomonas* sp. MACC-216 and *Chlorella* sp. MACC-360. Likewise, in *Scenedesmus obliquus*, the amount of chlorophyll a was shown to be decreased when the concentration of nitrate was increased from 12 mM to 20 mM [42]. Zarrinmehr et al. [23] showed the production of considerably large amounts of photosynthetic pigments in the presence of nitrate by *Isochrysis galbana*, in comparison to when there was no nitrogen present. They also observed a sharp drop in the amount of pigments when the concentration of nitrogen was increased from 72 mg L^−1^ to 288 mg L^−1^. On the other hand, *Neochloris oleoabundans* showed normal chlorophyll amounts at 15 mM and 20 mM, in comparison to lower concentrations of nitrate (3 mM, 5 mM and 10 mM), where a sharp drop in chlorophyll amount was observed [26]. It seems that the change in the amount of pigments in response to nitrate stress varies from algae to algae. In our study, total pigments of both microalgae were seemed to be affected by 10 mM and 15 mM concentrations of nitrate.

The effect of various nitrate concentrations on protein content was also investigated. Protein contents seemed to increase in both microalgae when the concentration of nitrate was increased from 5 mM to 10 mM and then it decreased when the concentration was further increased from 10 mM to 15 mM. Similar results were observed by Xie et al. [43], where they observed maximum protein production at 1.25 g L^−1^ nitrate, while, below and above this concentration, protein contents declined. Rückert and Giani [44] showed, in their studies, that the amount of protein was increased in the presence of nitrate, in comparison to when ammonium was used as nitrogen source. No continuous increase or decrease was observed in total carbohydrates when microalgae grown under different concentrations of nitrate were compared, probably because carbohydrate accumulation takes place under sulfur and nitrogen deprivation or nitrogen limitation as shown by previous studies [22,45,46].

Microalgae are known to accumulate neutral lipids, mainly in the form of triacylglycerols (TAGs) under environmental stress conditions. This lipid accumulation provides carbon and energy storage to microalgae to tolerate adverse environmental conditions. Through our study, it was observed that, while *Chlorella* sp. MACC-360 did not show any significant accumulation of lipids under the increasing concentration of nitrate, a significant increase in lipid accumulation was detected in *Chlamydomonas* sp. MACC-216 under the same conditions. In the case of *Chlorella* sp. MACC-360, we observed that results from BODIPY staining were not fully consistent with the lipid content results obtained from the phospho-vanillin reagent method; that is why lipid accumulation could not be considered significant in this microalga. It is interesting to note that, while *Chlamydomonas* sp. MACC-216 showed similar growth and nitrate removal in all of the media (TAP, TAP-M5, TAP-M10 and TAP-M15) by the 3rd day, they only showed lipid accumulation in the presence of 10 mM and 15 mM nitrate. The possible reason behind this could be the presence of non-utilized nitrate in the media and, probably, the accumulation of other nitrogenous compounds, such as nitrite and ammonia, produced after nitrate assimilation inside the microalgae, which act as stress factors. Similar to our results, the lipid content was increased when nitrate concentration was increased from 0 to 144 mg L^−1^ in *Isochrysis galbana* [23]. In contrast to our findings, previous studies have shown that the limitation of nitrogen increased the production of lipids in *Nannochloropsis oceanica*, *Nannochloropsis oculata* and *Chlorella vulgaris* [13,47,48,49]. 

Our study demonstrated the capability of the selected two microalgae to remove nitrate from concentrated synthetic wastewater. It was observed that *Chlamydomonas* sp. MACC-216 performed slightly better in removing nitrate than *Chlorella* sp. MACC-360 in SWW, despite showing slower growth. Previous studies have confirmed the capability of microalgae to remove nitrogen or its sources from wastewater [50,51,52,53,54]. McGaughy et al. [53] reported a 56% removal of nitrate from wastewater containing 9.3 mg L^−1^ nitrate by *Chlorella* sp., as measured on the 5th day of cultivation. Another study showed the removal of nitrate from secondary domestic wastewater treatment, where three different algae species, namely, LEM-IM 11, *Chlorella vulgaris* and *Botryococcus braunii*, removed 235 mg L^−1^, 284 mg L^−1^ and 311 mg L^−1^ of nitrate from the initial nitrate concentration of 390 mg L^−1^ by the 14th day of cultivation [50]. Through our study, it was observed that both *Chlamydomonas* sp. MACC-216 and *Chlorella* sp. MACC-360 have the capacity to grow and propagate in SWW containing a high concentration of nitrate and both microalgae performed well in removing a good portion (28–35%) of the initial nitrate content. 

Furthermore, the difference between the nitrate removal capacity of *Chlamydomonas* sp. MACC-216 and *Chlorella* sp. MACC-360 in TAP-M, compared to SWW, can probably be explained by the different composition of these media. Perhaps SWW has a significantly higher C/N value than TAP due to the extra carbon content of peptone and meat extract. It is likely that a heterotrophic metabolism is favoured by both algae in SWW and it represents a higher portion in the algal photoheterotrophic growth than when they are cultivated in TAP. Thus, a similar algal growth was achieved in SWW at a lower nitrate removal rate. 

## 5. Conclusions

Through this study, we sought out to determine the effects of varying concentrations of nitrate on two freshwater microalgae, *Chlamydomonas* sp. MACC-216 and *Chlorella* sp. MACC-360. Both microalgae were shown to have the capacity to remove nitrate with high efficiency. High nitrate concentrations led to lipid accumulation in *Chlamydomonas* sp. MACC-216, while protein and carbohydrate contents were not affected. We also revealed that high nitrate concentrations in SWW improved the growth of *Chlorella* sp. MACC-360 in comparison to *Chlamydomonas* sp. MACC-216. Both selected axenic green microalgae performed well in removing nitrate from synthetic wastewater. The nitrate removal capacity of these microalgae ought to be checked in real wastewater in future studies, where algal–bacterial interactions are expected to further increase the removal efficiency.

## Figures and Tables

**Figure 1 cells-10-02490-f001:**
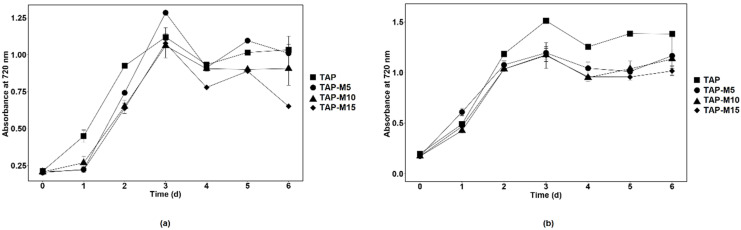
Growth of *Chlamydomonas* sp. MACC-216 (**a**) and *Chlorella* sp. MACC-360 (**b**) in TAP, TAP-M5, TAP-M10 and TAP-M15. Error bars represent standard deviations.

**Figure 2 cells-10-02490-f002:**
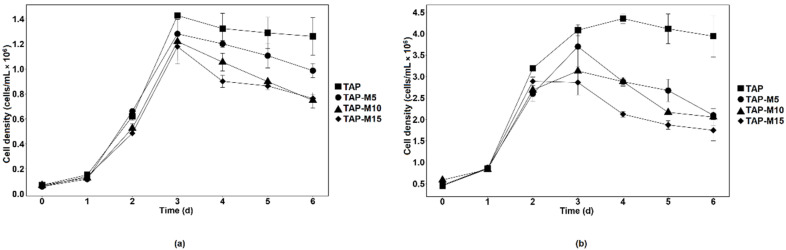
Cell density of *Chlamydomonas* sp. MACC-216 (**a**) and *Chlorella* sp. MACC-360 (**b**) in TAP, TAP-M5, TAP-M10 and TAP-M15. Error bars represent standard deviations.

**Figure 3 cells-10-02490-f003:**
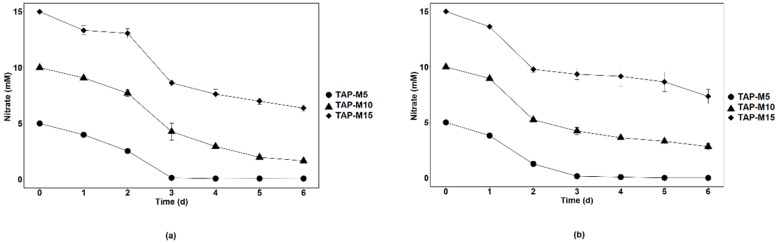
Nitrate removal by *Chlamydomonas* sp. MACC-216 (**a**) and *Chlorella* sp. MACC-360 (**b**) from media containing 5 mM (TAP-M5), 10 mM (TAP-M10) and 15 mM (TAP-M15) nitrate. Error bars represent standard deviations.

**Figure 4 cells-10-02490-f004:**
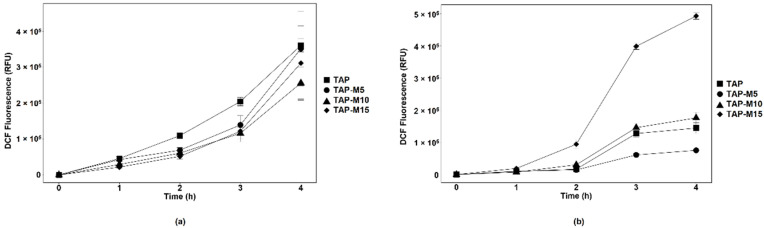
ROS production in *Chlamydomonas* sp. MACC-216 (**a**) and *Chlorella* sp. MACC-360 (**b**). Error bars represent standard deviations.

**Figure 5 cells-10-02490-f005:**
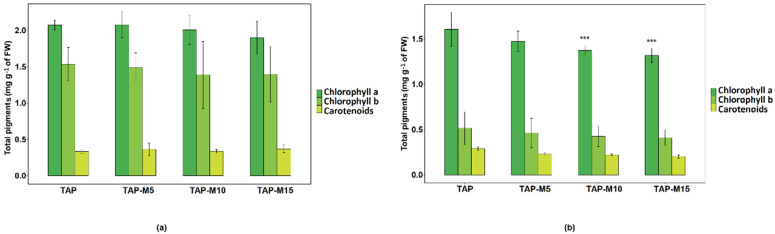
Total pigments of *Chlamydomonas* sp. MACC-216 (**a**) and *Chlorella* sp. MACC-360 (**b**) in TAP, TAP-M5, TAP-M10 and TAP-M15 media. Error bars represent standard deviations; asterisks (***) denote level of significance.

**Figure 6 cells-10-02490-f006:**
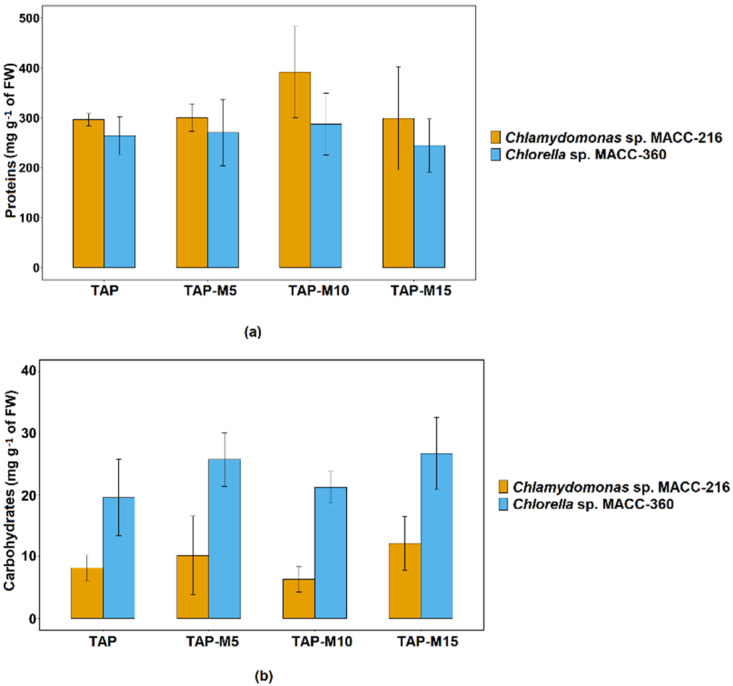
Total protein (**a**) and carbohydrate (**b**) contents of *Chlamydomonas* sp. MACC-216 and *Chlorella* sp. MACC-360. Error bars represent standard deviations.

**Figure 7 cells-10-02490-f007:**
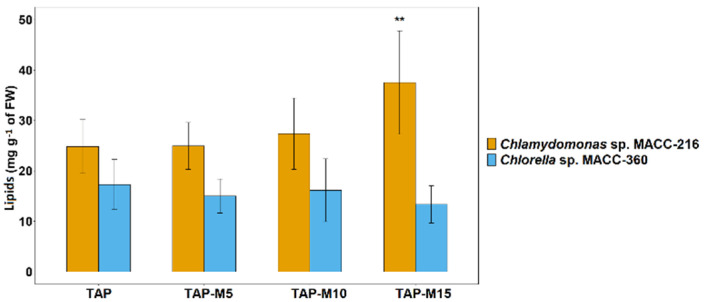
Total lipid contents of *Chlamydomonas* sp. MACC-216 and *Chlorella* sp. MACC-360 grown in TAP, TAP-M5, TAP-M10 and TAP-M15. Error bars represent standard deviations; asterisks (**) denote level of significance.

**Figure 8 cells-10-02490-f008:**
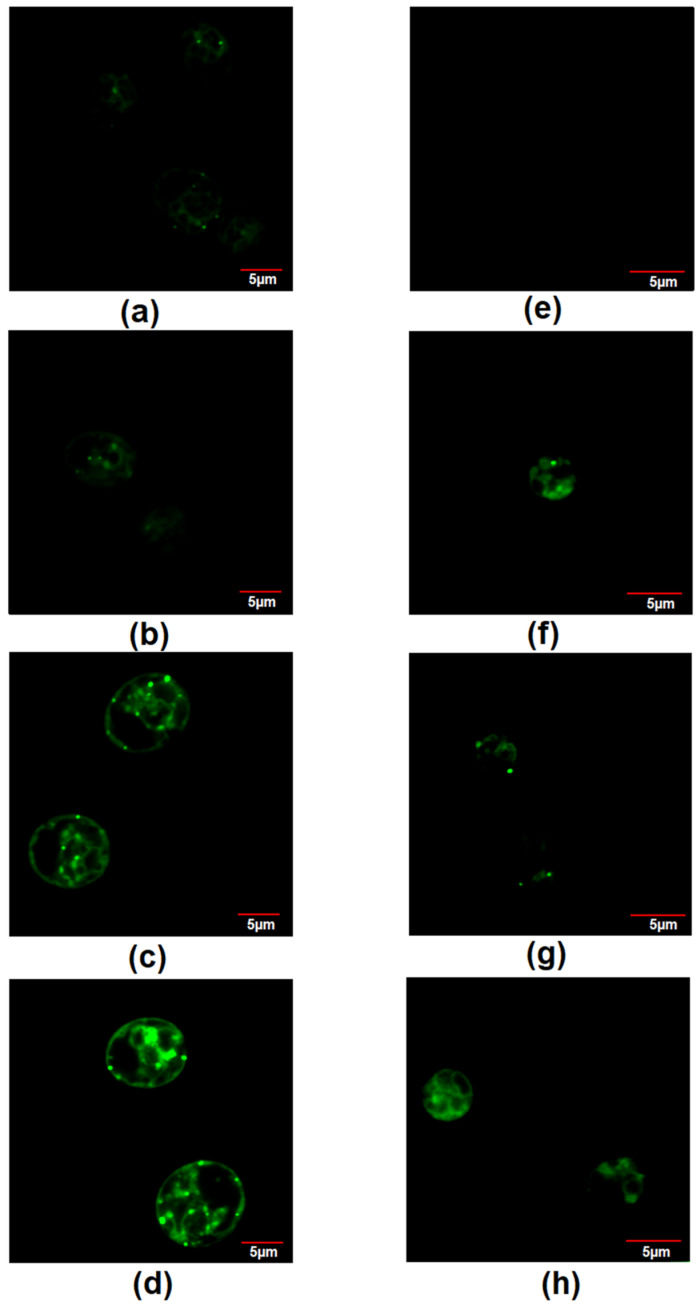
Staining of neutral lipids by BODIPY dye in *Chlamydomonas* sp. MACC-216 grown in TAP (**a**), TAP-M5 (**b**), TAP-M10 (**c**), TAP-M15 (**d**) and *Chlorella* sp. MACC-360 grown in TAP (**e**), TAP-M5 (**f**), TAP-M10 (**g**), TAP-M15 (**h**).

**Figure 9 cells-10-02490-f009:**
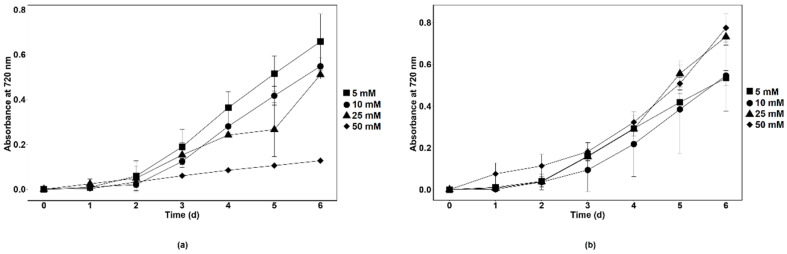
Growth of *Chlamydomonas* sp. MACC-216 (**a**) and *Chlorella* sp. MACC-360 (**b**) in SWW supplemented with 5 mM, 10 mM, 25 mM and 50 mM nitrate. Error bars represent standard deviations.

**Table 1 cells-10-02490-t001:** Growth parameters of *Chlamydomonas* sp. MACC-216 and *Chlorella* sp. MACC-360 in TAP, TAP-M5, TAP-M10 and TAP-M15 media. Values are represented as mean ± standard deviation.

Medium/Sample	*Chlamydomonas* sp. MACC-216				*Chlorella* sp. MACC-360			
	Number of Generations (n)	Mean Generation Time (g).	Specific Growth Rate Day^−1^ (µ)	Cell Size (µm)	Number of Generations (n)	Mean Generation Time (g)	Specific Growth Rate Day^−1^ (µ)	Cell Size (µm)
TAP	3.2 ± 0.0	0.6 ± 0.0	1.1 ± 0.0	24.2 ± 2.1	2.3 ± 0.1	0.9 ± 0.0	0.8 ± 0.0	12.9 ± 1.2
TAP-M5	3.5 ± 0.01	0.6 ± 0.0	1.2 ± 0.0	25.5 ± 2.8	2.1 ± 0.5	1 ± 0.2	0.7 ± 0.2	13.9 ± 2
TAP-M10	3.4 ± 0.5	0.6 ± 0.1	1.2 ± 0.1	25.1 ± 2.9	1.9 ± 0.0	1.1 ± 0.0	0.7 ± 0.0	15.2 ± 2.5
TAP-M15	2.9 ± 0.3	0.7 ± 0.1	1 ± 0.1	25.3 ± 3.3	1.7 ± 0.1	1.2 ± 0.1	0.6 ± 0.1	14.5 ± 1.7

**Table 2 cells-10-02490-t002:** Total nitrate removal by *Chlamydomonas* sp. MACC-216 and *Chlorella* sp. MACC-360 by 6th day. Values are represented as mean ± standard deviation.

Medium/Sample	Total Nitrate Removal					
	*Chlamydomonas* sp. MACC-216			*Chlorella* sp. MACC-360		
	mM	mg L^−1^	%	mM	mg L^−1^	%
TAP-M5	5 ± 0.0	425 ± 0.0	100 ± 0.0	5 ± 0.0	425 ± 0.0	100 ± 0.0
TAP-M10	8.4 ± 0.2	714 ± 18.2	84 ± 2.1	7.2 ± 0.3	613.8 ± 21.4	72.2 ± 2.5
TAP-M15	8 ± 0.3	683 ± 23.5	53.6 ± 1.8	7.7 ± 0.6	652.8 ± 54.3	51.2 ± 4.3

**Table 3 cells-10-02490-t003:** Nitrate removal rate of *Chlamydomonas* sp. MACC-216 and *Chlorella* sp. MACC-360 at three different time points. Values are represented as mean ± standard deviation.

Medium/Sample	Removal Rate (nmol Cell^−1^ h^−1^)					
	*Chlamydomonas* sp. MACC-216			*Chlorella* sp. MACC-360		
	3 h	6 h	9 h	3 h	6 h	9 h
TAP-M5	36.7 ± 2.8	61 ± 1.7	47.4 ± 1.3	6.6 ± 0.9	11.3 ± 0.0	10.7 ± 0.1
TAP-M10	30.1 ± 13.4	55.9 ± 2.8	53.9 ± 1.1	3.8 ± 0.1	11.4 ± 0.4	12.3 ± 0.0
TAP-M15	25.6 ± 5.2	55 ± 5.3	59 ± 3.6	10 ± 0.3	12.3 ± 0.01	13.3 ± 0.2

**Table 4 cells-10-02490-t004:** Total nitrate removal by *Chlamydomonas* sp. MACC-216 and *Chlorella* sp. MACC-360 in SWW supplemented with 5 mM, 10 mM, 25 mM and 50 mM nitrate by 6^th^ day. Values are represented as mean ± SD.

Medium/Sample	Total Nitrate Removal					
	*Chlamydomonas* sp. MACC-216			*Chlorella* sp. MACC-360		
	mM	mg L^−1^	%	mM	mg L^−1^	%
5 mM	1.8 ± 0.2	151.9 ± 20.2	35.8 ± 4.8	1.6 ± 0.1	134.2 ± 11.8	31.9 ± 2.8
10 mM	2.8 ± 0.2	237.1 ± 14.5	27.9 ± 1.7	3.7 ± 0.3	315.1± 25.8	37.1 ± 3.1
25 mM	9.5 ± 0.9	806.2 ± 75.4	38 ± 4	8.1 ± 0.8	692.3 ± 64.9	32.6 ± 3.1
50 mM	17.3 ± 0.8	1469.6 ± 66.5	34.6 ± 1.6	13.8 ± 0.8	1171 ± 65.5	27.6 ± 1.5

## Data Availability

The data presented in this study are available on request from the corresponding author.

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
