# Peer review of "Assessment of Nitrate Removal Capacity of Two Selected Eukaryotic Green Microalgae"

_cells, 2021, doi:10.3390/cells10092490_

Round 1
Reviewer 1 Report
This research work presents a study conducted with two eukaryotic green microalgae (Chlamydomonas sp. MACC-216 and 10 Chlorella sp. MACC-360) that were exposed to increasing nitrate concentrations. The influence of nitrate on the growth parameters of both microalgae, their nitrate removal ability under these conditions and their ROS production were evaluated. Moreover, authors analyzed how total pigments contents is affected by nitrate presence, as well as total protein, carbohydrate and lipid contents. Finally, the ability of both microalgae to efficiently remove nitrate from synthetic wastewater was studied.
From my point of view, the manuscript is properly written and the results are clearly explained, being easy to read and understand it. However, I believe that some considerations/corrections should be taken into account by authors before accepting this paper (minor revision):
- Figure 1 and figure 2 are redundant as some of the results showed in Fig 1 were repeated with the same conditions and shown in Fig 2. Maybe Fig 1 could be included as supplementary material as these experiments were only performed to select the suitable nitrate concentration for future experiments. In fact, in Figures 2 and 3, the peak of growth described for the third day is better observed.
- In Table 1: numbers are too close among them and it is difficult to read it properly. For Chlamydomonas, it is said that the highest specific growth rate was detected in TAP-M5 among all the other media (lines 263-264), but we can see in the table that in all the cases a similar specific growth rate was obtained. More differences are observed in the case of Chlorella
- Line 271: the title of this section should include also Chlorella strain
- From my point of view, discussion should be rewritten in order to organize well all the ideas that are exposed. For example, when it is mentioned in the text that after the 3rd day both microalgae were able to remove 100% of the nitrate from TAP-M5 (lines 410-412), it would be a good moment to explain how microalgae are able to transform nitrate and eliminate it from the media, that is explained then in the text (lines 423-432)
- It is mentioned in the discussion that an increase in the cell size of both microalgae was observed when nitrate concentration was also increased but these results are not shown in the manuscript and should be added to it.
- How can be explained the differences observed in both microalgae related to ROS production against nitrate presence? Why is Chlamydomonas less stressed than Chlorella? Maybe because of the differences observed in their pigments contents?
- Results obtained for Chlorella related to lipid accumulation in response to nitrate increase are confusing because with the BODIPY dye is observed an increase in lipid content in TAP-M media, is this true? In the discussion is confirmed that no significant differences were observed (lines 474-475). This point should be cleared up in the text.
Author Response
Reviewer 1
This research work presents a study conducted with two eukaryotic green microalgae (Chlamydomonas sp. MACC-216 and 10 Chlorella sp. MACC-360) that were exposed to increasing nitrate concentrations. The influence of nitrate on the growth parameters of both microalgae, their nitrate removal ability under these conditions and their ROS production were evaluated. Moreover, authors analyzed how total pigments contents is affected by nitrate presence, as well as total protein, carbohydrate and lipid contents. Finally, the ability of both microalgae to efficiently remove nitrate from synthetic wastewater was studied.
From my point of view, the manuscript is properly written and the results are clearly explained, being easy to read and understand it. However, I believe that some considerations/corrections should be taken into account by authors before accepting this paper (minor revision).
Q1. Figure 1 and figure 2 are redundant as some of the results showed in Fig 1 were repeated with the same conditions and shown in Fig 2. Maybe Fig 1 could be included as supplementary material as these experiments were only performed to select the suitable nitrate concentration for future experiments. In fact, in Figures 2 and 3, the peak of growth described for the third day is better observed.
Thank you for the valuable suggestion. We agree with your suggestion and decided to include Figure 1 in supplementary data.
Q2. In Table 1: numbers are too close among them and it is difficult to read it properly. For Chlamydomonas, it is said that the highest specific growth rate was detected in TAP-M5 among all the other media (lines 263-264), but we can see in the table that in all the cases a similar specific growth rate was obtained. More differences are observed in the case of Chlorella.
Thank you for making these suggestions. We have modified the values to point 1 decimal place in Table 1. As per your suggestion, we have also changed our statement regarding the specific growth rate of Chlamydomonas sp. MACC-216. We have now stated that- ’It was found that specific growth rate of Chlamydomonas sp. MACC-216 was similar among all of the four media, whereas, in the case of Chlorella sp. MACC-360, the highest specific growth rate was observed in TAP.’
Q3. Line 271: the title of this section should include also Chlorella strain
Thank you for your suggestion. We have made the change accordingly.
Q4. From my point of view, discussion should be rewritten in order to organize well all the ideas that are exposed. For example, when it is mentioned in the text that after the 3rd day both microalgae were able to remove 100% of the nitrate from TAP-M5 (lines 410-412), it would be a good moment to explain how microalgae are able to transform nitrate and eliminate it from the media, that is explained then in the text (lines 423-432)
Thank you for your valuable suggestion. We have made the changes accordingly.
Q5. It is mentioned in the discussion that an increase in the cell size of both microalgae was observed when nitrate concentration was also increased but these results are not shown in the manuscript and should be added to it.
Thank you for the comment. The cell size of both microalgae is mentioned in Table 1.
Q6. How can be explained the differences observed in both microalgae related to ROS production against nitrate presence? Why is Chlamydomonas less stressed than Chlorella? Maybe because of the differences observed in their pigments contents?
Reactive oxygen species are produced when microalgae are under stress. Various studies have shown the effects of high salinity and heavy metals on ROS production in microalgae but we could not find any study where there was a link explained between nitrate and ROS production. We are not sure why there is higher ROS production in Chlorella than Chlamydomonas. Few studies have shown that chlorophyll-a increased the production of ROS but in our study chlorophyll-a is decreasing with increasing concentration of nitrate in Chlorella sp. MACC-360, so we are not sure how it could lead to high ROS production. It is definitely a field that is worth investigating further.
Q7. Results obtained for Chlorella related to lipid accumulation in response to nitrate increase are confusing because with the BODIPY dye is observed an increase in lipid content in TAP-M media, is this true? In the discussion is confirmed that no significant differences were observed (lines 474-475). This point should be cleared up in the text.
With BODIPY dye, we observed that few cells of Chlorella sp. MACC-360 (~5-10 among 100 cells) started to show BODIPY fluorescence with increasing concentration of nitrate but when we further investigated the lipid content by phospho-vanillin reagent assay, we did not find any significant difference between lipid contents of microalgae grown in four different media (TAP, TAP-M5, TAP-M10, TAP-M15). That is why we mentioned in discussion that there was no significant difference.
Reviewer 2 Report
The manuscript (Cells-1340213) of Vaishali Rani and Gergely Maróti
titled “Nitrate Bioremediation Potential of Two Selected Eukaryotic Green Microalgae” before publication in Cells needs some minor corrections.
Comments:
Page 1
Line 32 should be “NO3– L-1” instead of “NO3 L-1”
Page 2
Line 73 should be “MgSO4•7H2O” instead of “MgSO4•H2O” and in whole MS
Line 76 lack of acetic acid concentration
Line 86 check the unit of light intensity
Line 92 should be “720 nm” instead of “720nm”
Page 3
Line 121 should be “2M NaOH“ instead of “2N NaOH“
Page 4
Line 176 should be “50 μL“ instead of “50μL“
Line 183 should be “4000 rpm“ instead of “4000rpm“
Line 184 should be “3 mL” instead of “3 ml”
Line 189 should be “2 mL” instead of “2 ml”
Page 5
Line 193 should be “4000 rpm” instead of “4000rpm”
Line 196 should be “100 mL” instead of “100 ml” ; should be “400 mL” instead of “400 ml”
Line 200 should be “100 μgL” instead of “100 μgl”
Line 217 should be “1 L” instead of “1L” ; there is no information about the conductivity of
distilled water
Line 226 the given unit (50 μE) requires clarification
Page 7
Line 269 Table 1. is unclear
Page 8
Line 285 should be “6th” instead of “6th”
Page 9
Table 3. should be “3 h” instead of “3h”; should be “6 h” instead of “6h”; should be “9 h” instead of “9h”
Line 317 should be “chlorophyll a” instead of “chlorophyll-a”
Line 319 should be “chlorophyll b” instead of “chlorophyll-b”
Page 15
Line 449 should be “chlorophyll a, b” instead of “chlorophyll-a, b”
Line 455 should be “chlorophyll a” instead of “chlorophyll-a”
Line 458 should be “5 mM” instead of “5mM”
Line 465 should be “5 mM to 10 mM” instead of “5mM to 10mM”
Line 466 should be “10 mM to 15 mM” instead of “10mM to 15mM”
Author Response
Reviewer 2
The manuscript (Cells-1340213) of Vaishali Rani and Gergely Maróti titled “Nitrate Bioremediation Potential of Two Selected Eukaryotic Green Microalgae” before publication in Cells needs some minor corrections.
Q1. Line 32 should be “NO3– L-1” instead of “NO3 L-1”
Thank you for your suggestion. We have made the change accordingly.
Q2. Line 73 should be “MgSO4•7H2O” instead of “MgSO4•H2O” and in whole MS
Thank you for your suggestion. We have made the change accordingly.
Q3. Line 76 lack of acetic acid concentration
Thank you for your comment. We have provided the concentration of acetic acid.
Q4. Line 86 check the unit of light intensity
Thank you for your comment. We have modified the unit of light intensity.
Q5. Line 92 should be “720 nm” instead of “720nm”
Thank you for your suggestion. We have made the change accordingly.
Q6. Line 121 should be “2M NaOH“ instead of “2N NaOH“
Thank you for your suggestion. We have made the change accordingly.
Q7. Line 176 should be “50 μL“ instead of “50μL“
Thank you for your suggestion. We have made the change accordingly.
Q8. Line 183 should be “4000 rpm“ instead of “4000rpm“
Thank you for your suggestion. We have made the change accordingly.
Q9. Line 184 should be “3 mL” instead of “3 ml”
Thank you for your suggestion. We have made the change accordingly.
Q10. Line 189 should be “2 mL” instead of “2 ml”
Thank you for your suggestion. We have made the change accordingly.
Q11. Line 193 should be “4000 rpm” instead of “4000rpm”
Thank you for your suggestion. We have made the change accordingly.
Q12. Line 196 should be “100 mL” instead of “100 ml” ; should be “400 mL” instead of “400 ml”
Thank you for your suggestion. We have made the changes accordingly.
Q13. Line 200 should be “100 μgL” instead of “100 μgl”
Thank you for your suggestion. We have made the change accordingly.
Q14. Line 217 should be “1 L” instead of “1L” ; there is no information about the conductivity of distilled water
Thank you for your suggestion and comment. We have made the change from “1L” to “1 L”. The electrical conductivity of the distilled water that we used was 5 µS cm-1. We did not mention it in our manuscript because this distilled water was only used for the preparation of synthetic wastewater, so according to us, conductivity should not matter there.
Q15. Line 226 the given unit (50 μE) requires clarification
Thank you for your comment. We have modified the unit of light intensity to 50 µmol m−2 s−1.
Q16. Line 269 Table 1. is unclear
Thank you for your comment. We have modified Table 1.
Q17. Line 285 should be “6th” instead of “6th”
Thank you for your suggestion. We have made the change accordingly.
Q18. Table 3. should be “3 h” instead of “3h”; should be “6 h” instead of “6h”; should be “9 h” instead of “9h”
Thank you for your suggestion. We have made the changes accordingly.
Q19. Line 317 should be “chlorophyll a” instead of “chlorophyll-a”
Thank you for your suggestion. We have made the change accordingly.
Q20. Line 319 should be “chlorophyll b” instead of “chlorophyll-b”
Thank you for your suggestion. We have made the change accordingly.
Q21. Line 449 should be “chlorophyll a, b” instead of “chlorophyll-a, b”
Thank you for your suggestion. We have made the change accordingly.
Q22. Line 455 should be “chlorophyll a” instead of “chlorophyll-a”
Thank you for your suggestion. We have made the change accordingly.
Q23. Line 458 should be “5 mM” instead of “5mM”
Thank you for your suggestion. We have made the change accordingly.
Q24. Line 465 should be “5 mM to 10 mM” instead of “5mM to 10mM”
Thank you for your suggestion. We have made the change accordingly.
Q25. Line 466 should be “10 mM to 15 mM” instead of “10mM to 15mM”
Thank you for your suggestion. We have made the change accordingly.
Reviewer 3 Report
The paper reports a comparative nitrate removal from synthetic wastewater (SWW) using two microalgae strains and the effect of different nitrate concentrations on growth and physiology of
Chlamydomonas sp. MACC-216 and 10 Chlorella sp. MACC-360. This subject is thoroughly studied, so the novelty of the present study is not clear. Although the range of nitrate concentrations used is very wide (from 5 mM to a really high 100 mM), the results presented for different concentrations are confusing. Why only 5-15 mM concentrations were tested for growth parameters, ROS, pigments, sugars and lipid storage, when the nitrate removal from SWW was studied at 5-50 mM? Especially for Chlorella that grew better in SWW with higher (25 and 50 mM) nitrate concentrations.
Specific comments
- A title should be reformulated to avoid ‘nitrate bioremediation’.
- Abstract should be revised to show clearly was new finding were obtained. All conflicting conclusions should be corrected, for example ‘both microalgae were grown under three different concentrations of sodium nitrate i.e, 5 mM (TAP-M5), 10 mM (TAP-M10) and 15 mM (TAP-M15)’ and then ‘…while Chlorella MACC-360 preferred SWW containing 25 mM and 50 mM nitrate for growth’. Also ‘Both microalgae were able to fully remove nitrate from the TAP-M5 medium by 3rd day’ and then ‘Both microalgae were able to remove approximately one-third of nitrate by the 6th day…’. Only most essential results should be presented. It is suggested to remove the sample names (TAP-M5, TAP-M10 and TAP-M15) as TAP and TAP-M are not specified here.
- The introduction should provide more studies on the main subject, either on nitrate removal from WW using microalgae or effects of different nitrate concentrations on studied physiological parameters of microalgae.
- L60-67 The source of strains and nutrient medium should be in the Materials and methods.
- MACC – the culture collection web site should be provided, and the phrase that these strains were maintained at the Institute of Plant Biology, Biological Research Centre, Szeged, Hungary should be removed. Usually, the collection strains are identified to the species level, so the current taxonomic status of strains should be refined.
- L237-239 This is confusing as TAP contains ammonium salts (NH4Cl and (NH4)6MoO3,) and Tris base (NH2C(CH2OH)3), so nitrate cannot be ‘a sole nitrogen source’.
- It is not clear what is difference between Fig. 1 and Fig. 2, only in the presence of NH4Cl in the TAP medium? Also Fig. 2 and Fig. 3 illustrate the same data obtained by two different methods (OD by microplate reader and cell count by LUNA)? LUNA counting was used with DAPI staining? It should be specified.
- Table 1 is difficult to read and it contains some errors, e.g specific growth rate for Chlamydomonas should be 0.118 or 1.118?. What is ‘Mean generation days (g)’? Perhaps mean generation time (mean doubling time), days. Also check for other parameters.
- Since data in Fig. 4 and Table 2 are calculated from the same experimental result and largely duplicate each other, it is suggested to present the nitrate removal efficiency in % in the Table 2.
- Table 3, since SD and mean values are in the same numerical order, they should be presented as e.g. 37 ± 3 rather than 36.7 ± 2.828. This comment is also addressed to Table 1 and 4.
- Different results on nitrate removal from TAP-M and SWW should be discussed in relation with the optimal C:N ratio for different nitrate concentrations. A revealed non-linear dependence of nitrate removal activity from the growth of microalgae should be also discussed, perhaps in relation with their physiology (protein and lipid accumulation).
Author Response
Reviewer 3
The paper reports a comparative nitrate removal from synthetic wastewater (SWW) using two microalgae strains and the effect of different nitrate concentrations on growth and physiology of Chlamydomonas sp. MACC-216 and 10 Chlorella sp. MACC-360. This subject is thoroughly studied, so the novelty of the present study is not clear. Although the range of nitrate concentrations used is very wide (from 5 mM to a really high 100 mM), the results presented for different concentrations are confusing.
Q1. Why only 5-15 mM concentrations were tested for growth parameters, ROS, pigments, sugars and lipid storage, when the nitrate removal from SWW was studied at 5-50 mM? Especially for Chlorella that grew better in SWW with higher (25 and 50 mM) nitrate concentrations.
We first started our study with modified TAP media. There we checked the growth of Chlamydomonas sp. MACC-216 and Chlorella sp. MACC-360 on varying concentrations of sodium nitrate from 1 mM to 100 mM (Supplementary Figure 1) to select three nitrate concentrations for further experiments and we decided to continue our study with 5 mM, 10 mM and 15 mM nitrate concentrations. After checking all the parameters in modified TAP, we wanted to see the microalgae’s performance in synthetic wastewater as well. The original recipe of synthetic wastewater consisted of 50 mM sodium nitrate so we decided to do three more dilutions, that is how we generated SWW with four different concentrations of sodium nitrate (5mM, 10 mM, 25 mM and 50 mM) in total.
Q2. A title should be reformulated to avoid ‘nitrate bioremediation’.
The title has been changed, the term ‘nitrate bioremediation’ was removed. The new title is simply ‘Assessment of Nitrate Removal Capacity of Two Selected Eukaryotic Green Microalgae’.
Q3. Abstract should be revised to show clearly was new finding were obtained. All conflicting conclusions should be corrected, for example ‘both microalgae were grown under three different concentrations of sodium nitrate i.e, 5 mM (TAP-M5), 10 mM (TAP-M10) and 15 mM (TAP-M15)’ and then ‘…while Chlorella MACC-360 preferred SWW containing 25 mM and 50 mM nitrate for growth’. Also ‘Both microalgae were able to fully remove nitrate from the TAP-M5 medium by 3rd day’ and then ‘Both microalgae were able to remove approximately one-third of nitrate by the 6th day…’. Only most essential results should be presented. It is suggested to remove the sample names (TAP-M5, TAP-M10 and TAP-M15) as TAP and TAP-M are not specified here.
Thank you for your valuable suggestion. We have made the changes accordingly. We decided to keep the sample names i.e., TAP-M5, TAP-M10 and TAP-M15 and specified in the abstract that first we grew both microalgae in modified tris-acetate-phsophate medium (TAP-M).
Q4. The introduction should provide more studies on the main subject, either on nitrate removal from WW using microalgae or effects of different nitrate concentrations on studied physiological parameters of microalgae.
Thank you for the valuable suggestion. As per your suggestion, we have modified the introduction and provided more studies on nitrate removal from wastewater and also included few more studies on the effects of nitrate on microalgae.
Q5. L60-67 The source of strains and nutrient medium should be in the Materials and methods.
Thank you for your suggestion. We have included source of strains in Materials and methods section. We still believe that it is important to note in the introduction that TAP and SWW media were used in this study. Details of these media are provided in the Materials and Methods section.
Q6. MACC – the culture collection web site should be provided, and the phrase that these strains were maintained at the Institute of Plant Biology, Biological Research Centre, Szeged, Hungary should be removed. Usually, the collection strains are identified to the species level, so the current taxonomic status of strains should be refined.
The MACC is not a publicly available collection, although it is open for research collaboration. A short description of the collection is available at the website of the Szechenyi University: https://plantbio.sze.hu/en_GB/mosonmagyarovar-algal-culture-collection. The phrase that “these strains were maintained at the Institute of Plant Biology, Biological Research Centre, Szeged, Hungary” was removed. Most members of the MACC were collected in the 1980’s and 90’s, 30-40 years ago. Most of the isolates were characterized simply by their morphology using light microscope. Of course, we have performed molecular taxonomy on the selected isolates. Even so, we have found that the general taxonomy of Chlorophyta was far from clear, various studies used a number of different markers to determine the phylogenetic positions (16S rDNA, 18S rDNA, tufA, rbcL, etc.). Thus, we think that only genus level can be responsibly described at the moment for most of the isolates.
Q7. L237-239 This is confusing as TAP contains ammonium salts (NH4Cl and (NH4)6MoO3,) and Tris base (NH2C(CH2OH)3), so nitrate cannot be ‘a sole nitrogen source’.
Thank you for the comment. Yes, normal TAP contains ammonium salts. But in our modified TAP medium, there is no NH4Cl, we substituted it with NaNO3. We also replaced (NH4)6Mo7O24 · 4H2O with Na2MoO4 · 2H2O in this modified TAP medium. We have mentioned it now in the Materials and methods section. As for Tris base, we tried to grow both microalgae with Tris base as only nitrogen source just to check whether it can serve as a nitrogen source, but microalgae did not grow at all. That’s why we decided to mention nitrate as the sole nitrogen source. Now, we have re-phrased the line to as “sole nitrogen source for the growth.”
Q8. It is not clear what is difference between Fig. 1 and Fig. 2, only in the presence of NH4Cl in the TAP medium? Also Fig. 2 and Fig. 3 illustrate the same data obtained by two different methods (OD by microplate reader and cell count by LUNA)? LUNA counting was used with DAPI staining? It should be specified.
Figure 1 (now Supplementary Figure1 )depicts the growth of both microalgae over a wide range of sodium nitrate concentrations in modified TAP medium. It was done to select three particular concentrations of nitrate for further experiments, whereas Figure 2 (now Figure 1) depicts the growth of both microalgae in normal TAP and modified TAP with 5mM, 10 mM, 15 mM concentration of nitrate. Figure 2 (now Figure 1) and Figure 3 (now Figure 2 ) illustrates the cell growth and cell number of both microalgae. For cell growth, we took OD or absorbance by microplate reader and for cell number LUNA cell counter was used. We did not use any dye for cell counting. The counting was done on the basis of autofluorescence of the microlalgae. As per your suggestion, we have included this information in the materials and methods section.
Q9. Table 1 is difficult to read and it contains some errors, e.g specific growth rate for Chlamydomonas should be 0.118 or 1.118?. What is ‘Mean generation days (g)’? Perhaps mean generation time (mean doubling time), days. Also check for other parameters.
We have corrected Table 1. By ‘g’ we meant to denote mean generation time. We have corrected that.
Q10. Since data in Fig. 4 and Table 2 are calculated from the same experimental result and largely duplicate each other, it is suggested to present the nitrate removal efficiency in % in the Table 2.
Thank you for your suggestion. We have modified the table and included nitrate removal efficiency in %.
Q11. Table 3, since SD and mean values are in the same numerical order, they should be presented as e.g. 37 ± 3 rather than 36.7 ± 2.828. This comment is also addressed to Table 1 and 4.
Thank you for your comment. We have modified values in all of the tables up to point 1 decimal place. We decided to keep values with point 1 decimal place for precision.
Q12. Different results on nitrate removal from TAP-M and SWW should be discussed in relation with the optimal C:N ratio for different nitrate concentrations. A revealed non-linear dependence of nitrate removal activity from the growth of microalgae should be also discussed, perhaps in relation with their physiology (protein and lipid accumulation).
The difference between the nitrate removal capacity of Chlamydomonas sp. MACC-216 and Chlorella sp. MACC-360 in TAP-M compared to SWW can probably be explained by the different composition of these media. SWW has a significantly higher C/N value than TAP due to the extra carbon content of peptone and meat extract. It is likely that heterotrophic metabolism is favoured by both algae in SWW and it represents higher portion in the algal photoheterotrophic growth compared to when cultivated in TAP. Thus, similar algal growth was achieved in SWW at lower nitrate removal rate.
Round 2
Reviewer 3 Report
The comments were attained and the article was corrected accordingly. Only English grammar and style should be improved in some places.